# Sex and Gender Differences in Kidney Cancer: Clinical and Experimental Evidence

**DOI:** 10.3390/cancers13184588

**Published:** 2021-09-13

**Authors:** Anna Julie Peired, Riccardo Campi, Maria Lucia Angelotti, Giulia Antonelli, Carolina Conte, Elena Lazzeri, Francesca Becherucci, Linda Calistri, Sergio Serni, Paola Romagnani

**Affiliations:** 1Department of Experimental and Clinical Biomedical Sciences “Mario Serio”, University of Florence, Viale Morgagni 50, 50134 Florence, Italy; marialucia.angelotti@unifi.it (M.L.A.); giulia.antonelli@unifi.it (G.A.); carolina.conte@unifi.it (C.C.); elena.lazzeri@unifi.it (E.L.); linda.calistri@unifi.it (L.C.); paola.romagnani@unifi.it (P.R.); 2Unit of Urological Robotic Surgery and Renal Transplantation, Careggi Hospital, University of Florence, 50134 Florence, Italy; riccardo.campi@unifi.it (R.C.); sergio.serni@unifi.it (S.S.); 3Department of Experimental and Clinical Medicine, University of Florence, Viale Morgagni 50, 50134 Florence, Italy; 4Nephrology and Dialysis Unit, Meyer Children’s University Hospital, Viale Pieraccini 24, 50139 Florence, Italy; francesca.becherucci@meyer.it

**Keywords:** kidney cancer, renal cell carcinoma, gender, incidence, risk factors, outcomes, hormone signaling axis, sex

## Abstract

**Simple Summary:**

Kidney cancer is a frequent malignant tumor that accounts for approximately 5% of all cancer incidences. It affects both males and females, but males are twice as likely to develop kidney cancer than females. Evidence shows that this discrepancy takes root in individual differences, such as genetics or pathologies that affect the patient. It is then reflected in the clinical characteristics of the tumors, as males have larger and more aggressive tumors. Understanding the sex- and gender-based differences in kidney cancer is essential to be able to offer patients individualized medicine that would better cover their needs in terms of prevention, diagnosis and treatment.

**Abstract:**

Sex and gender disparities have been reported for different types of non-reproductive cancers. Males are two times more likely to develop kidney cancer than females and have a higher death rate. These differences can be explained by looking at genetics and genomics, as well as other risk factors such as hypertension and obesity, lifestyle, and female sex hormones. Examination of the hormonal signaling pathways bring further insights into sex-related differences. Sex and gender-based disparities can be observed at the diagnostic, histological and treatment levels, leading to significant outcome difference. This review summarizes the current knowledge about sex and gender-related differences in the clinical presentation of patients with kidney cancer and the possible biological mechanisms that could explain these observations. Underlying sex-based differences may contribute to the development of sex-specific prognostic and diagnostic tools and the improvement of personalized therapies.

## 1. Introduction

Sex and gender are two terms that are often used interchangeably, which has been at the origin of misunderstandings and misconceptions in recent years. In this review, we will use the term sex when referring to biological and physiological characteristics and gender when referring to characteristics that are socially constructed, as defined by the World Health Organization (WHO) (https://www.who.int/health-topics/gender#tab=tab_1, accessed on 9 September 2021).

Sex-related differences in incidence and mortality have been reported for a wide range of cancers [1]. Analysis of the United States Cancer Statistics (USCS) public use database from 2001 to 2016 revealed that males have a higher incidence and worse survival outcomes than females for most cancer sites, including bladder, kidney, colorectum, liver, esophagus, head and neck, brain, skin, and blood [1]. However, while oncology experts are calling for more attention and investigation, gender is often overlooked in the clinical management of the patient [2].

Worldwide, kidney cancer is the sixth most frequent cancer in males and the 10th in females, representing 5% and 3% of all new cases, respectively [3]. It is the fifth non-reproductive cancer in terms of male to female incidence ratio, with a 2.0 ratio, following esophagus, bladder, liver and head and neck [4]. In this review, we will first report the sex-based disparities in cancer incidence, followed by the impact of genetics and genomics on this sex imbalance and the risk factors that contribute to these differences. Then, we will examine the hormone signaling pathways in kidney cancer, to provide some answers to the sex disparity conundrum. Next, we will explore the tumor characteristics, treatment and prevention in male and female patients, as well as the surgical, oncological and functional outcomes (Figure 1). Finally, we will discuss how sex-related specificities should be given major importance in the clinical management of patients with kidney cancer.

## 2. Incidence

Epidemiological studies report that males have a twofold greater risk of developing kidney cancer in their lifetime than females. An analysis of worldwide cancer incidence data for the years 1978–2007 indicated a 2:1 male/female case incidence ratio that was constant by age, year and region [5]. The analysis of the Surveillance, Epidemiology, and End Result (SEER) database for the years 2001–2016 show similar rate in males, with age-adjusted incidence rate double that of female in the US [6]. Data from global vital registries revealed a 23.04% increase in age-standardized incidence in kidney cancer between 1990 and 2013 worldwide, which corresponded to a 31.19% increase in males, compared to 8.79% in females [7]. According to the Global Cancer Observatory: CANCER TODAY (https://gco.iarc.fr/today/home, accessed on 9 September 2021), the estimated age-standardized incidence rates in 2020 for kidney cancer worldwide was 4.6 for both sexes, 6.1 for males and 3.2 for females. The Cancer Statistics Center of the American Cancer Society (https://cancerstatisticscenter.cancer.org/, accessed on 9 September 2021) reported incidence rates (average annual rate per 100,000, age adjusted to the 2000 US standard population) of 16.9 for both sexes, 22.9 for males and 11.7 for females, for the period 2013–2017. The European Cancer Information System (https://ecis.jrc.ec.europa.eu/, accessed on 9 September 2021) registered for 2020 incidence rates (average annual rate per 100,000, age adjusted) for Europe of 18.4 for both sexes, 25.9 for males and 12.5 for females.

## 3. Genetics and Genomics

People with a family history of cancer have an increased risk of developing kidney cancer [8] and the heritability of kidney cancer is estimated to be around 38% (https://www.cancer.gov/types/kidney/hp/renal-cell-carcinoma-genetics, accessed on 9 September 2021), suggesting that genetics is an important determinant of kidney tumorigenesis. However, the great majority of hereditary and sporadic kidney cancers do not present genetic sexual dimorphisms. As an exception, in tuberous sclerosis complex (mutation in TSC1, 9q34.13 or TSC2, 16p13.3), angiomyolipomas represent the typical lesions and are more frequent in females. From a genetic point of view, angiomyolipomas frequently show loss of heterozygosity in TSC2 or TSC1. Fewer than five percent of patients with tuberous sclerosis complex develop renal cell carcinoma (RCC) [9], which occurs primarily in females [10]. In hereditary papillary renal carcinoma (mutation in MET, 7q31.2) and hereditary leiomyomatosis and renal cell cancer (mutation in fumarate hydratase, FH, 1q43) the main histotype of kidney cancer is papillary RCC (pRCC), type 1 and type 2, respectively. The male: female ratio of hereditary papillary renal carcinoma is 2.4:1 [11], but the determinant of this diversity is still unknown. Although symptomatic uterine tumors are frequently the cause of hereditary leiomyomatosis and renal cell cancer diagnosis, there are no data about a sex prevalence of RCC in this syndrome. Interestingly, lipid peroxidation that is increased after gynecological surgery (e.g., hysterectomy, that has been reported as a risk factor for kidney cancer) can induce DNA damage and promote mutations in protoncogenes and tumor suppressor genes [8].

Genomic studies showed that the genome of males and females differ by about 1.4% of the overall content [12], potentially explaining differences in disease risk and association, as well as in prognosis. Escape from X-chromosome inactivation (“lyonization”) of tumor suppressor genes is a mechanism hypothesized to contribute to the overall reduced incidence of cancer observed in females [13], including clear cell RCC (ccRCC). Caceres et al., provided additional evidence to support this mechanism, revealing that several genes of the non-recombinant region whose chromosome X homologs showed loss-of-function mutations, co-occurred with extreme downregulation of chromosome Y (EDY) during cancer [14]. In their study, kidney cancer was associated with EDY (OR 20.1) despite low participant numbers (12 case patients and 17 control patients), suggesting that EDY could represent a signature of cancer risk in males and the functional mediator of the reported association between the mosaic loss of chromosome Y and cancer [14]. Of note, loss of Y chromosome has been reported in 47% of tumors in a FISH (fluorescence in-situ hybridization) analysis of 1252 male renal tumors, and it was prevalent in pRCC (77%) followed by chromophobe RCC (60%), oncocytoma (51%), ccRCC (39%) and clear cell papillary RCC (19%) [15]. Recently, a meta-analysis of Xp11.2 translocation RCC, which involves gene translocation and fusion in X chromosome, was performed to assess the difference of morbidity between male and female [16]. The authors reported that the incidence in females was higher than that of males (OR 2.84), suggesting that the sex difference observed is based on the fact that the number of X chromosomes in female is double that of male [16]. Insights into sex-specific genomic differences in kidney cancer have been provided by gene expression studies, especially for RCC. A meta-analysis of gene expression profiles of more than 6000 genes in ccRCC identified about 350 autosomal genes regulating critical tumor-associated biological cell functions that were differentially expressed between males and females [17]. Male tumors showed overexpression of immune and inflammation-associated genes, suggesting tumor infiltration by immune cells. On the other hand, female tumors overexpressed genes of catabolic-related processes [17]. Recently, a sex-specific genome-wide association study (GWAS) performed in over 8000 patients with RCC provided evidence for sex-specific associations for two known RCC genetic loci (DPF3, 14q24.2 and EPAS1, 2p21, associated with female and male sex, respectively), further supporting the role of genetic susceptibility in determining RCC sexual dimorphisms [18]. In a follow-up study, sex-specific gene expression was found both in normal and in tumor tissues, involving autosomal and sex genes (e.g., genes subject to X-inactivation) [19]. In general, males showed skewed expression of genes potentially relevant for RCC cancerogenesis. Interestingly, these differences were even more evident in RCC samples and involved expression of genes associated with non-malignant tumor-predisposing metabolic disorders with male prevalence, as well as expression of genes regulating immune and metabolic pathways potentially relevant for disease determination and progression [19]. The data also indicate that the inflammatory tumor environment, that is an important hallmark of cancer known to affect proliferation, survival, metastasis and alter response to therapeutic agents, is sex-specific [19]. The results presented by Brannon et al., and by Laskar et al. [17,19] highlight the existence of sex differences with regard to immune response in RCC patients, a phenomenon already reported in autoimmune and infectious diseases [20]. The role of inflammation in the surveillance and treatment of RCC and the implications of inflammatory cells within tumors have been known for many years [21,22], and attempts have been made to treat several types of cancer through regulation of the immune response [23]. However, in light of this data, particular attention should be made to sex while treating kidney cancer patients with immune checkpoint inhibitors. Tan et al., investigated the relationship between sex differences in fatty acid-binding protein7 (FABP7) and POU class 3 homeobox 2 (BRN2) expression in RCC and found that FABP7 was more expressed in males while BRN2 was highly expressed in females, and showed that expression levels of FABP7 and BRN2 had prognostic value in females [24]. Shish et al., analysed the molecular signature of patients affected with metastatic RCC and identified two genes (CTHRC1, BCL2L14) that were overexpressed and one gene (AGBL4) that was underexpressed in females [25]. While CTHRC1 has been showed to favor RCC progression, there is no evidence on the role of the other two genes in RCC, suggesting the need for additional studies [24]. These observations pave the way for further investigations to understand the biological difference of RCC between sexes.

## 4. Other Risk Factors

The main risk factors for kidney cancer include obesity, diabetes, hypertension, smoking, chronic and acute kidney injury, drugs and nephrotoxic agents [26]. Interestingly, several studies suggest a gender-specific impact of certain risk factors. Additionally, the sex hormones seem to weight on the risk of kidney cancer.

### 4.1. Smoking and Occupational Hazard

It has been suggested that the higher incidence of kidney cancer in males could originate from the higher frequency of smokers and from increased occupational hazard in the male population [27]. While the prevalence of current tobacco product use among U.S. adults has declined over the past several decades, it remained higher among males (26.2%) than among females (15.7%) [28]. Additionally, smoking seemed to affect more males than females. About 22.4% of renal cell carcinoma- (RCC-) related deaths among males were attributed to smoking in 2013 worldwide, but only 8.9% in developed countries and 3.1% in developing countries for females [7]. While exposure to cancerogenous substances is a known risk factor to RCC development in both sexes, males face an increased risk of occupational hazards. For example, an epidemiologic study shows a statistically significant increased risk of RCC among the highest users of 2,4,5-trichlorophenoxyacetic acid (2,4,5-T), an herbicide [29]. The authors enrolled 57,310 individuals seeking licenses to apply for restricted-use pesticides in Iowa and North Carolina between 1993 and 1997, of whom only about 2.7% were female [29]. Similarly, in a study presenting a dose-dependent association between RCC and exposure to metalworking fluids among autoworkers, the cohort consisted of 33,421 individuals, of whom 86.5% were males [30].

### 4.2. Obesity and Hypertension

Obesity is one of the well-established risk factors for kidney cancer [31]. Meta-analyses indicate a strong association between an elevated body mass index (BMI) and RCC, in particular in females [32]: the relative risk (RR) of developing renal cancer for patients with a BMI ≥ 35 was 2.47 for males and 2.59 for females. In a systematic review and meta-analysis of prospective observational studies, the authors reported a slightly stronger association between a 5 kg/m^2^ increase in BMI and renal cancers in females (RR 1.34) than in males (RR 1.24) [32]. In a case-control study on a population of about 500 consecutive Chinese female patients with pathologically confirmed RCC, obese pre-menopausal females had an increased risk of RCC (OR 1.67), after adjusting for age, hypertension and diabetes, while no such positive association was observed among the subjects as a whole, suggesting a role for estrogens [33].

Hypertension is an independent risk factor for RCC. Analysis of the VITamin And Lifestyle (VITAL) cohort of 77,719 individuals aged 50 to 76 living in a 13-county area of western Washington State showed that hypertension was independently associated with RCC risk (hazard ratio, HR 1.70) [34]. Hypertension is more frequent in males than in females [35], and hypertensive males are 1.32 times more likely to develop RCC [36]. While the use of antihypertensive medications has been significantly associated with an increased risk of kidney cancer, there was no association with gender subgroups [37]. Importantly, hypertension is a known symptom and risk factor for kidney diseases, frequently leading to chronic kidney disease (CKD) with a more drastic worsening of kidney function in males [38], and as such represent a risk factors for kidney cancer [39].

### 4.3. Female Sex Hormones

While most epidemiological studies report that incidence rates in males are consistently higher than in females, some studies show that sex differences lessen with age [40], suggesting a possible role for female hormones.

#### 4.3.1. Hysterectomy and Oophorectomy

Hysterectomy, that is the surgical removal of the uterus, is one of the most common procedures to treat benign and malignant gynecological conditions worldwide. In a 2014 systematic review and meta-analysis of 7 cohort and 6 case-control studies, the authors found a sum of squared residuals (SSR) for hysterectomy and kidney cancer of 1.29, irrespectively of age at hysterectomy, time since the procedure and model adjustment for BMI, smoking status and hypertension [41]. Of note, not all studies, taken individually, reported an association [41]. A recent retrospective cohort study supported these findings, as they found an HR for hysterectomy and kidney cancer of 1.32 [42]. A study based on the Women’s Health Initiative (WHI), a large prospective study in the US involving about 145,000 females, reported that hysterectomy associated with an increased risk of RCC (HR 1.28) [43]. Interestingly, younger (≤40 years old) and older (≥55 years old) females were more at risk [43]. These results echo a 2010 study, based on Swedish registry databases, that reported an increased risk of RCC in females younger than 44-year-old (HR 2.36) [44]. Among the possible reasons for the association between hysterectomy and risk of RCC, residual cofounding by indication for hysterectomy cannot be excluded [43]. Other potential reasons include renal damage or pelvic anatomy changes caused by surgery, and an increased likelihood to undergo later abdominal imaging leading to incidental detection of smaller tumors [43]. The surgical removal of one or both ovaries, or oophorectomy, is another widespread gynecological procedure. In particular, bilateral salpingo-oophorectomy (BSO) is frequently performed at the time of hysterectomy. Wilson et al., showed that hysterectomy-BSO were associated with a higher risk of kidney cancer (HR 1.29) [42]. The WHI study, however, does not associate oophorectomy with an increased risk of RCC [43].

#### 4.3.2. Reproductive and Hormonal Factors

Evidence shows that high parity is associated with an increased risk of RCC. A Swedish population-based study reported that parous females had an increased risk of RCC compared to nulliparous females (odds ratio (OR) 1.42), with a major risk for females having 5 or more births (OR 1.91) [45]. Consistently, a Canadian cohort study, reported that parous females were more at risk than nulliparous ones (HR 1.78), and even more so for females of high parity (HR 2.41) [46]. In agreement with the previous reports, Lee et al., found that compared with 1 or 2 childbirths, the multivariate relative risks were 1.75 for 4 childbirths and 1.50 for high parity [47]. Physiological and hormonal changes in pregnant females, such as increased estrogen levels, renal hyperfiltration and weight gain, could account for the association of parity with kidney cancer [46]. Regarding the link between hormone intake and kidney cancer risk, Liu et al., found that oral contraceptive use may reduce the risk of kidney cancer, in particular for long-term users [48]. Conversely, others found no association between RCC and parity, age and type of menopause, use and duration of oral contraceptive and type and duration of postmenopausal hormone use [8,49,50]. In agreement with these studies, Peila et al., recently showed no association of endogenous sex hormone levels (circulating estradiol, testosterone and sex hormone binding globulin (SHBG)) with the risk of kidney cancer in the UK Biobank cohort [51].

Taken together, these results do not exclude a connection between female sex hormones and RCC.

## 5. The Hormonal Signaling Axis

While epidemiological studies have established the existence of differences between male and female kidney cancer patients, several studies are lifting the veil on how sex influences kidney biology and cancer development, with evidence supporting a role for hormonal signaling in kidney cancer (Figure 2).

### 5.1. The Androgen Signaling Axis

The androgen signaling axis is composed of the androgen receptor (AR) and the androgen synthesis pathways. AR expression was observed in 15 to 55% of RCC samples [52,53,54,55]. Some studies showed that AR expression was significantly associated with lower pathological stage and favorable outcome [53,55]. Conversely, others reported that AR levels were significantly higher in patients with high-stage tumors and correlated with poor prognosis [54,56]. Interestingly, in the study by Ha et al., about 25% of patients were females, demonstrating that the androgen pathway has a role regardless of sex [54]. As expected, the data regarding the level of expression in normal vs. tumoral tissue or male vs. female RCC patients is also conflicting [52,53,54,55,56]. The discordant results of these studies could derive from methodological differences. Indeed, evaluation of AR expression with immunohistochemical staining, quantitative PCR or at protein level via western blot might not reflect a functional AR in RCC patients [53].

Recent reports tried to identify the pathways involved in AR-mediated RCC formation. He et al., demonstrated that cells isolated from normal human kidney transfected with AR were able to develop more and larger colonies in soft agar when treated with a carcinogen [57]. In addition, AR showed to promote RCC cell migration and invasion [57]. The treatment of RCC cells with hypoxia-induced factor 2α (HIF2α) and vascular endothelial growth factor (VEGF) inhibitors, showed that RCC progression could be suppressed, suggesting an important key role of HIF2α/VEGF signaling in RCC progression. The tumor suppressor von Hippel-Lindau (*VHL*) gene is one of the most important suppressors of the HIF2α/VEGF pathway and 45% of metastatic RCC present the mutant or lost *VHL* gene [58], which might lead to constitutively activated HIF-VEGF signals to induce the RCC progression [59,60]. He et al., identified the AR as another player to induce the HIF/VEGF pathway in RCC, which might be a potential target for RCC therapy. Targeting AR with its degradation enhancer, in RCC preclinical models, they obtained a suppression of RCC tumor progression [57]. Angiogenesis is a well-known mechanism involved in the physiopathology of RCC [60]. Guan et al., demonstrated that AR induced vascular endothelial cells (ECs) proliferation and recruitment by modulating protein kinase B (AKT) → nuclear factor kappa-light-chain-enhancer of activated B cells (NF-κB) → C-X-C Motif Chemokine Ligand 5 (CXCL5) signaling [61]. Phosphoinositide 3-kinase (PI3K)/AKT pathway plays a key role in RCC proliferation and invasion, cancer stem cell maintenance and angiogenesis in the tumor [62]. In Guan’s study, the authors identified PI3K/AKT signaling as downstream target of AR, which increased CXCL5 expression. This increase in CXCL5 in RCC cells induced EC recruitment into the tumor microenvironment. A recent study established that HOX transcript antisense RNA (HOTAIR) and AR created a feedback loop, promoted the transcription of GLI Family Zinc Finger 2 (GLI2) and increased the expression of GLI2 downstream genes, including VEGFA and Platelet Derived Growth Factor Subunit A (PDGFA). Accordingly, the authors proved that HOTAIR and AR were involved in tumor angiogenesis and cancer stemness maintenance in vitro and in vivo via the Hedgehog-GLI2 signaling pathway [63]. Recently, other AR-mediated mechanisms involved in RCC progression were described. A key enzyme of the urea cycle, argininosuccinate synthase 1 (ASS1), was found to have an important role in the proliferation and progression of some types of tumors [64,65]. Wang et al., reported that AR expression is negatively correlated with ASS1 expression, regulating the competing endogenous RNA activity of its pseudogene ASS1P3. This mechanism could induce RCC proliferation and progression. Increased expression of ASS1P3 could result in reducing cell proliferation, representing a potential therapeutic approach for advanced RCC [66].

Pak et al., evaluated the effect of dihydrotestosterone (DHT) on the proliferation of RCC cells in relation to AR status. They found that DHT promoted cell proliferation through STAT5 (signal transducer and activator of transcription 5) activation in RCC cells regardless of the AR status. AR inhibition showed antitumor activity in RCC cell lines [67].

To assess the potential benefit of targeting androgen signaling in AR-positive RCC, Lee et al., implanted AR-positive RCC cells in mice, and treated them with enzalutamide, an AR inhibitor, or abiraterone acetate (AA), a CYP17A1 (Cytochrome P450 17A1) inhibitor that suppresses the production of androgens [68]. They observed a significant reduction of the tumor. Follow-up studies showed that lysine-specific histone demethylase 1 (LSD1), an epigenetic co-regulator of AR, knocked-down in kidney cancer cells led to a slower growth and decreased migration ability of the cells in vitro and in a xenograft mouse model [69]. These effects were enhanced by co-treatment with enzalutamide. From the results of these studies, a clinical trial has been set up to study the effect of targeting AR in RCC in a clinical setting. The BARE trial (Blockade of Androgens in renal cell carcinoma using Enzalutamide, NCT02885649, www.clinicaltrials.gov, accessed on 9 September 2021) is a phase 0 ongoing clinical trial which investigates the effects of enzalutamide treatment on tumor growth prior to surgical resection.

In summary, all these reports provide novel insight to understand the role of androgen and AR in the pathogenesis of kidney cancer and identify AR as a potential therapeutic target to suppress RCC progression and improve patient’s outcomes.

### 5.2. The Estrogen Signalling Axis

The biological functions of estrogens are traditionally mediated by binding to one of two estrogen receptors (ERs), ERα and ERβ, and subsequent regulation of downstream genes. In RCC, the level and frequency of ER expression is highly variable, with a higher frequency of ERβ than ERα, and there are very limited and contrasting data concerning the role of estrogenic signals in the tumorigenesis and development of RCC [52,70,71,72,73]. Previous studies demonstrated that estrogen, through ERβ signaling, suppressed the proliferation, migration and invasion of RCC cells and increased RCC apoptosis [74,75,76]. Analysis of the molecular mechanisms revealed that estrogen reduced the expression of downstream genes, such as AKT, extracellular signal-regulated kinase (ERK) and Janus kinase (JAK) signaling pathways and increased the apoptotic genes (BH3 interacting-domain death agonist (Bid), Caspase-3, Caspase-8 and Caspase-9), through ERβ activation, suggesting that this receptor could have an inhibitory effect in RCC [75,76]. In contrast, other studies demonstrated that ERβ could be an oncogene and play promoting roles in RCC progression. Results from human clinical data indicated that there was higher ERβ expression in tumors at later stages or higher grades and that the increased ERβ expression was associated with a worse survival and a lower disease-free survival for RCC patients [77,78,79,80].

However how ERβ regulates RCC progression and the underlying mechanisms remain under investigation. Recent studies using in vitro RCC cells or in vivo mouse models suggested that ERβ could modulate the functions of circular RNA-ATP2B1, and long non-coding RNA-HOTAIR and consequently antagonize several miRNAs to upregulate oncogenes that promote RCC progression [79,80]. In addition, ERβ expression promoted RCC cell growth and increased RCC metastasis by altering the expression of transforming growth factor β1 (TGFβ1)/Small mother against decapentaplegic homolog 3 (SMAD3) signals [78]. Targeting ERβ/TGFβ1/SMAD3 signals with FDA-approved anti-estrogen Faslodex or with an ERβ selective antagonist significantly reduced RCC tumor growth and invasion [78]. Other reports suggested that infiltrated immune cells could alter ERβ to promote RCC progression [81,82]. In particular, the recruited T cells increased ERβ expression and enhanced RCC cell invasion via altering the ERβ → Disabled homolog 2-interacting protein (DAB2IP) signals [81]. Moreover, infiltrating neutrophils promoted RCC progression via VEGFα/HIF2α and ERβ signals [82].

Genetic polymorphism of ERα in the kidney also seems to be involved in the development of renal cancer [83]. Recently ERα was identified as a novel target of tumor suppressor von Hippel-Lindau protein (pVHL) E3 ligase [84]. Overexpression of pVHL promoted the ubiquitin-mediated degradation of ERα, whereas downregulation of pVHL increased ERα expression. Blocking of ERα using Faslodex suppressed the proliferation of VHL-deficient RCC cells [84]. Interestingly an alternatively spliced variant of ERα was identified, called ERα36. The expression of this receptor correlated with the progression, treatment resistance and poor prognosis of many types of cancers [85,86]. ERα36 was located in the plasma membrane and cytoplasm, rather than in the nuclei as ERα, transduced rapid non-genomic estrogenic signaling to stimulate cell growth and inhibited the transcription activities of both ERα and ERβ [87]. High expression of ERα36 in the cytoplasm and on the membrane correlated with bad prognostic factors in RCC: poor disease-free survival, larger tumor size and late clinical stage [85,88].

Recently, a seven-transmembrane receptor G-protein coupled estrogen receptor (GPER) was suggested to mediate the rapid non-genomic signal of estrogen [89]. Increasing evidence showed that GPER contributed to cell migration and proliferation of certain cancers [90]. In renal cancer, GPER was highly expressed in RCC cell lines and promoted the migration and invasion of RCC cells via the PI3K/AKT/MMP-9 signals [91].

Renal tumors can be experimentally induced in the golden Syrian hamsters, by sustained treatment with synthetic (diethylstilbestrol) or natural (estradiol) estrogens. These renal tumor models have been useful to study the hormonal carcinogenesis and suggested the involvement of ERs and xenoestrogen in the etiology of renal cancer [92,93,94]. Sex differences were observed in the incidence of estrogen-induced renal tumor in these experimental models. Indeed, treatment of both intact and gonadectomized hamsters with estrogens led to high incidences of renal tumors in males and a lower incidence rate in females [95,96].

In vitro studies suggested that potentially reactive intermediates of estrogen may be the causative factors of the experimental nephron-carcinogenesis, contributing to a severe oxidative stress in renal cells during prolonged exposure to estrogen [97,98].

Taken together, the role of estrogens in kidney cancer remain controversial, with some studies suggesting an oncogenic role and others suggesting a protective role. The variation of ERα/ERβ ratio as well as the different levels and subcellular localization of their splice variants seem to contribute to the complexity of estrogen actions. Thus, further investigations are required to understand the role of estrogens in the pathogenesis and how estrogens contributed to the sex disparities in kidney cancers.

### 5.3. The Progesterone Signaling Axis

Expression of progesterone receptors (PRs) in carcinomatous kidney tissue suggests a potential implication of progesterone axis in developing RCC [99]. Expression of PRs was investigated in several tumor types. Whereas their expression in ccRCC was low, they resulted to be valid routine diagnostic markers to distinguish oncocytoma from chromophobe RCC. Indeed, the former shows a higher expression of PRs than the chromophobe RCC [52,100]. Angiomyolipoma, a benign kidney tumor that increases in size following pregnancy or oral contraceptive therapy, also showed an increase expression of PRs, suggesting a potential role in this type of tumor as well. Two hypotheses have been made about distinct mechanisms involved in RCC formation: 1. A direct role in which estrogen exposure increases RCC; 2. An indirect role in which prolonged estrogen treatment increases progesterone-specific binding levels increasing the risk of developing RCC. An increased expression of PRs is a favorable prognostic marker in RCC developing. Several clinical observations demonstrate the efficacy of progesterone-based therapy, employed after nephrectomy. Of note, patients with hormone-dependent tumors with an increase in either ER and/or PR in absence of metastasis, benefit from the progestational therapy, showing a favorable outcome [101,102,103].

## 6. Diagnosis

The classic triad of flank pain, gross haematuria and palpable abdominal mass is rare (6–10%) among contemporary patients with renal cancer [104]. More than 50% of RCC are detected incidentally by non-invasive imaging performed for a variety of non-specific symptoms and/or other abdominal concerns. In this regard, previous studies have shown that male patients may be significantly younger than female patients at the time of diagnosis, raising the hypothesis of a potential protective role of female hormones at younger ages [40,105].

While the most recent European Association of Urology (EAU) Guidelines on RCC did not mention any potential sex-related difference in renal cancer diagnosis [104], there is evidence that female patients may experience consistently delayed evaluations for haematuria and urinary tract infections (UTI) as well as longer diagnostic intervals for cancer diagnosis [106]. In particular, the time interval between presentation of symptoms and hospital referral was found to be longer in females with bladder or renal cancer as compared to males, regardless of the presence of haematuria. This suggests a potential sex-related bias in the interpretation of haematuria among physicians [107]. As such, an English primary care audit survey showed that females required more prereferral consultations for renal cancer diagnosis as compared to males; this finding was confirmed at multivariable analysis adjusting for age, haematuria status and use of primary care-led investigations. Such evidence points to a potentially clinically relevant discrepancy in diagnostic delays between sexes [107]. A recent systematic review reinforced this finding [106].

It should also be noted that, while their contribution to the incidence of RCC is still unclear, urinary tract infections have been described as a modifiable risk factor for the development of RCC [108,109,110]. This risk has been shown to be potentially modulated by sex (i.e., increased in males) [109]. The recent discovery of a distinct urinary tract microbiome [111] provides a foundation for novel fields of research exploring the potential role of microbiome and RCC pathogenesis [110]. It should be noted, however, that evidence on the impact of sex on a potential diagnostic delay for RCC is still controversial [112].

In a population-based study using data from the SEER database [113], tumors in females were significantly smaller (5.9 cm vs. 6.1 cm) and with a lower pathological grade as compared to their male counterparts. Moreover, the incidence of regional or metastatic spread of RCC was lower among females. This finding is supported by other studies using data from the CORONA database [114], that highlighted a more unfavorable tumor grading (grade 3–4 tumors; 22.8% vs. 17.8%) as well as a higher incidence of distant metastasis (7.6% vs. 5.8%) among males. However, in this study, the maximum tumor diameter and pTN stages were similar in both sexes [114]. Previous evidence is in line with these findings [12,115,116]. In both studies, the sex-related difference in tumor stage at diagnosis (with females being diagnosed at an earlier stage than males) was associated with a more frequent incidental diagnosis of localized RCC in females. Lastly, the proportion of female patients presenting with stage IV RCC appears to be significantly lower than males [117], as previously reported by other groups [118].

Yet, disentangling the ultimate role of sex from that of other factors (such as the access and pattern of imaging examinations) is complex in light of the currently available evidence. In fact, several studies on this topic may not provide a contemporary view on renal cancer diagnostic pathways [119] and did not take into account how different diagnostic patterns might have contributed to the observed rates of renal cancer diagnoses among males and females. For example, in the study by Beisland et al., the rate of females in the incidental detected tumor group was higher than in the symptomatic group, which may be explained by an earlier diagnosis with lower stage at diagnosis in female patients [120], raising concerns regarding the potential impact of women’s increased health conscious attitude on the pattern of RCC presentation.

## 7. Histology

While there is growing evidence showing that sex may play a role in cancer incidence, progression and response to therapy [121], robust data on sex-related differences in the distribution of the main RCC histologic subtypes are currently lacking. The most recent WHO classification of renal neoplasms account for more than 50 entities and provisional entities [122]. Cystic nephromas and solitary fibrous tumors (two benign entities) seem more prevalent in females, perhaps owing to hormonal influences on the development and progression of these tumors [12]. On the contrary, males are more likely to be diagnosed with oncocytoma than females [104]. In addition, the prevalence of renal angiomyolipomas is significantly higher in females as compared to males [104]. Concerning RCC, while the most common histological subtype in both sexes remains ccRCC, translocation RCC (TRCC) Xp11.2 and TRCC t(6;11) occur more commonly in younger females, while tubulocystic RCC (a very rare tumor entity) is significantly more prevalent among males [104].

Importantly, in line with previous studies, in a landmark analysis of a large US population-based database including 18,060 patients undergoing partial nephrectomy between 2007 and 2014, female sex (in addition to old age and performance of computer tomography (CT) imaging only as preoperative imaging modality) was found to be a significant predictor of final benign pathology after surgery (reported in 30.9% of cases) [123]. These findings were confirmed by other studies [124] and by a recent meta-analysis by Pierorazio et al. [125,126].

While only a few studies have explored sex-related distribution of histological features of RCC, available data suggest that there might be relevant differences in the distribution of RCC subtypes across sexes. Namely, females tend to develop less likely pRCC while are more frequently diagnosed with chromophobe RCC than males. This difference may be clinically important, as it might partially explain the more favorable outcomes of RCC among females (considering the less aggressive behavior of chromophobe RCC) [12,105,116]. A recent Japanese retrospective study confirmed these findings [127]. Similarly, the sex-stratified analysis by Bhindi et al., found that the percentages of both malignant histology as well as aggressive histology were consistently higher for males [115].

Notably, the question as to whether genetic and molecular profiles of RCC vary according to sex is still open to debate [125].

## 8. Treatment

To date, female sex is not considered as a relevant factor influencing treatment decision-making (surveillance vs. treatment; treatment modality; and surgical approach) in patients with localized renal masses [104,128]. In a collaborative review, Chandrasekar et al., outline the complexity of the current decision-making schemes for localized RCC, given the need to balance several and often competing patient- and tumor-specific clinical variables [129]. Nevertheless, previous studies have assessed the potential impact of sex on treatment-related outcomes such as access to surgery, surgical approach and surgical outcomes [104]. For example, in a population-based study, Marchioni et al., found that male patients were more likely to be offered a surgical management as compared with female patients [130]. Other studies reported that females were less likely to undergo nephron-sparing surgery as compared to males [112,125,131,132,133,134], despite having lower odds of pathologic upstaging and high-grade disease [135]. Such sex disparity seems to be particularly evident among African-American females [136]. Similarly, in the setting of metastatic RCC, Patel et al., found that females were less likely to undergo a cytoreductive nephrectomy as compared to their male counterparts [137]. In a population-based cohort study using data from the National Cancer Database, females with localized renal masses were treated more aggressively than males, with consequent significantly higher risks of overtreatment [138]. Despite these data, there is no mention of any potential sex-related differences in access to care and type of treatment pattern by the latest international Guidelines [104]. In patients that underwent robotic partial nephrectomy, in addition to tumor complexity, male sex was associated with a significantly higher risk of not achieving optimal surgical outcomes [139,140].

Sex has been showed to have an influence on cancer treatment, in particular on immunotherapy [141,142]. However, a recent systematic review and meta-analysis of immunotherapy response to advanced kidney cancer established that sex did not influence overall survival (OS) or progression-free survival (PFS) [143]. The analysis of the International Metastatic RCC Database Consortium (IMDC) confirmed this result [144], although the absence of statistical significance of the interaction test between patients’ sex and magnitude of nivolumab efficacy could be due to the small sample [145]. A prospective observational cohort study in metastatic non-small cell lung cancer, melanoma and RCC patients revealed a significant effect of sex, body surface area and serum albumin on nivolumab clearance [146]. Lung cancer patients with progressive disease had a significantly higher drug clearance, and a similar trend was observed in RCC patient, suggesting that future prospective studies on the pharmacokinetics of nivolumab in kidney cancer patients could help improve clinical management [146]. In the metastatic clear-cell renal cancer setting, six phase 3 randomized controlled trials have investigated immune checkpoint blockade (through programmed cell death protein 1 (PD-1) or its ligand PD-L1), with the treatment consisting of PD-1/PD-L1 inhibition in combination with therapy targeting CTLA-4 signaling or VEGF [147,148,149,150,151,152]. The comparator arm was sunitinib in all the studies. Based on the results of these trials, the updated (2021) European Association of Urology (EAU) Guidelines recommend immune checkpoint inhibitors (ICI) as first line treatment across all risk classes as first-line treatment of metastatic clear-cell renal cancer [153]. In fact, four ICI combinations with proven overall survival (OS) benefit represent the new standard of care for these patients [153]. Importantly, pembrolizumab plus lenvatinib, nivolumab plus cabozantinib and pembrolizumab plus axitinib showed benefit irrespective of International Metastatic RCC Database Consortium (IMDC) risk group and PD-L1 status. A recent systematic review aiming to compare the available ICI-based strategies for systemic treatment of metastatic renal cancer concluded that pembrolizumab plus axitinib seemed to be the most efficacious first-line agents, while nivolumab plus ipilimumab showed the most favorable efficacy–tolerability equilibrium [154]. Notably, the potential impact of sex on oncologic outcomes in patients with metastatic (clear-cell) renal cancer treated with contemporary ICI-based systemic therapy is still controversial and not entirely investigated. In a recent systematic review and meta-analysis [141], both males and females with metastatic renal cell carcinoma had an OS and PFS benefit with immunotherapy, but no statistically significant difference between sexes was reported, raising concerns regarding the effect of sample size or inherent differences in the etiology of cancer [142]. In the subgroup analysis of OS among International Metastatic Renal Cell Carcinoma Database Consortium (IMDC) intermediate- and poor-risk patients in the CheckMate 214 (NCT02231749) cohort [147], the survival advantage of nivolumab plus ipilimumab versus sunitinib was more pronounced among female patients. The same finding was reported for pembrolizumab plus axitinib vs. sunitinib in a sub-group analysis of overall survival in the intention-to-treat population of the KEYNOTE-426 (NCT02853331) trial [149]. In the latter study, the PFS advantage of ICIs vs. sunitinib in the intention-to-treat population was also more pronounced for female patients. On the contrary, in a similar sub-group analysis of PFS among patients with PD-L1–Positive Tumors in the JAVELIN Renal 101 (NCT02684006) trial [148], a significant survival advantage of avelumab plus axitinib vs. sunitinib was reported only for male patients. Similarly, in the CheckMate 9ER (NCT03141177) trial [152], a more pronounced PFS advantage of nivolumab plus cabozantinib vs. sunitinib was noted for male patients. The same finding was noted in the sub-group analysis of PFS in the CLEAR (NCT02811861) trial comparing Lenvatinib+Pembrolizumab vs. sunitinib [150]. Taken together, these findings should prompt more extensive research on the potential impact of sex on oncologic outcomes of metastatic (clear-cell) renal cancer patients treated with ICIs in the future.

Despite the limited number of studies, evidence shows that males and females present differences in response to radiotherapy, as demonstrated in oesophageal squamous cell carcinoma [155] and rectal cancer [156], suggesting a possible benefit from personalized therapy [157,158]. However, so far no such difference has been reported in kidney cancer.

Overall, the current evidence is limited by the relatively low number of studies investigating sex-related differences in the effectiveness of specific therapeutic interventions for patients with kidney cancer. In addition, several studies published to date did not take into consideration all major potential confounders (related to patients, tumors, and healthcare contexts/practices) that could significantly influence the association between sex and oncologic or functional outcomes after surgical or ablative treatments for kidney cancer. Such confounders include tumor-specific factors (stage, grade and aggressiveness), patient-related factors (comorbidities, frailty and life expectancy), as well as provider-related factors (type of treating Centre, team’s experience and skills, availability of multidisciplinary tumor boards, etc.). This lack of granular data hinders meaningful comparison of available studies, making the design of prospective studies assessing the impact of sex on clinically relevant patient outcomes a key priority for clinicians and researchers.

## 9. Outcomes

Overall, the prognosis of RCC is better in females than in males. Data from global vital registries revealed a 11.3% mortality decrease in females between 1990 and 2013 worldwide and a 9.9% increase in males [7]. Other cancer registry analyses support this observation [159]. In a multicenter, retrospective study, May et al., showed that female sex positively influenced disease-specific survival (HR 0.75) and overall survival (HR 0.80), but sex addition in multivariable models did not significantly gain predictive accuracies [114]. Rampersaud et al., reported that females <59-year-old presented a better overall survival in comparison to males, suggesting a role for the hormones in RCC development [160]. Surprisingly, in metastatic RCC patients, female sex had a significant independent association with death (HR 1.19) [137]. Mancini et al., reported that disease recurrence seems not to be different among males and females, with the latter showing a more favorable prognosis after recurrence [105]. Wood et al., showed that male sex, a solitary kidney at partial nephrectomy, positive surgical margins, multiple tumors and higher nephrometry score and pathological stage increased the risk of local tumor bed recurrence [161]. In a retrospective study from 1957 to 1995 in Japan, of 768 RCC patients that underwent nephrectomy, females also had better survival after tumor recurrence than males (*p* = 0.007) [117]. Fukushima et al., also associated female sex with lower Fuhrman grade and recurrence-free survival (at 5 years, 92%, vs. 87% in males) in ccRCC patients [162].

Few studies associate sex with functional outcome of the kidney. Dagenais et al., showed that male sex associated with a decreased kidney function after robot-assisted partial nephrectomy in a single-center study on 647 patients [163]. Zabell et al., found male sex as a predictor of CKD 5 years after surgery as well as of nonrenal cancer mortality 10 years after renal cancer surgery [164]. A recent single-center retrospective study showed that male sex associated with acute kidney injury (AKI) and lower kidney function following NSS [165].

Of note, females suffer more psychological distress following kidney cancer surgery than males. In a retrospective study on non-metastatic RCC patients, females presented higher psychological distress sub-score (PDSS) at all times from diagnosis to treatment [166]. In a complementary study, Decat Bergerot et al., reported that younger, female patients had an increased fear of cancer recurrence [167].

## 10. Discussion

Several risk factors for kidney cancer have been established for years [26], such as obesity, diabetes, hypertension, smoking and CKD, while others have been described very recently, like AKI [39]. It is to be expected that in the next few years more risk factors will be revealed and included in the standard clinical practice. To date, the latest EAU Guidelines to prevent kidney cancer focus on the three most established modifiable risk factors, that is cigarette smoking, obesity and hypertension, calling for a healthier lifestyle. Currently, EAU does not mention any gender-related differences with regard to the distribution of RCC-related risk factors or potential actions aimed at reducing their prevalence and impact in the general population [104]. Yet, a few studies include male gender as a distinct, unmodifiable, risk factor for RCC development [168]. Not only is RCC incidence increasing worldwide and mortality rates are high but also a high proportion of individuals are asymptomatic at diagnosis, suggesting a possible benefit for tailored screening programs for RCC among high-risk individuals [168]. To that aim, individual patient risk-stratification must be established, based on established risk factors as well as gender, to improve screening efficiency and minimize harms by identifying the most suitable candidates for such programs [169].

Regarding kidney cancer diagnosis in females, a major importance should be given to recurrent or refractory urinary tract symptoms and hematuria, which could otherwise lead to delayed bladder or kidney cancer detection. In particular, Zhou et al., showed that UTI and nephrolithiasis before cancer diagnosis were associated with diagnostic delay, in females more than in males [106]. Additionally, because these symptoms might be mistaken for pregnancy-related disorders, routine antepartum examinations during physiological pregnancies fail to increase the incidental finding of kidney renal tumors [170].

Furthermore, while sex-related differences in kidney cancer characteristics and outcomes are considerable, they are often not taken into consideration in the treatment of the disease. In particular, females present sex-related differential pharmacokinetics compared to males, due to their usually lower bodyweight and organ size, higher percentage of body fat, lower glomerular filtration rate and different gastric motility [171]. Dosing regimens must be adapted to the sex of the patient to improve the balance between efficacy and toxicity for drugs with significant pharmacokinetic differences. Furthermore, while radiotherapy and chemotherapy offer survival advantages to females for several types of cancer, cardiac toxicity occurs at lower dose [4]. The scarcity of studies reporting the sex disparity in cancer treatment limits our ability to offer precision medicine to kidney cancer patients. Sex-specific medicine is quickly becoming a discipline of major importance, which urgently requires considerable basic science research and clinical research.

Despite recent advances in our understanding of the mechanisms responsible for sex-related differences in the pathophysiology of diseases, sex bias in experimental design remains a major hurdle in clinical research as well as in basic science [172]. It can be overt, inadvertent, situational, financial or ignorant, but always results in conclusions that might apply to only one sex. For example, historically, male rats and mice have been the subject of choice to study a variety of diseases, under the wrong assumption that female fluctuating hormones would make their results more variable and hard to interpret [173]. To remediate to this shortcoming, in 2015 the NIH (National Institute of Health) in the U.S. implemented a policy mandating consideration of “sex as a biological variable” and requiring researchers to include both sexes in their vertebrate animal and human studies [174]. This is an important step to correct sex-biased research, but many more years will have to pass before the flawed notion that males are a standard from which females might deviate disappears.

## 11. Conclusions

Sex and gender disparities have been reported for kidney cancer, with a higher incidence and worse outcome in males than in females. Differences in risk factors, genetics, sex hormones and tumor characteristics all contribute to this trend. Further research will elucidate these differences and facilitate personalized patient care and perhaps support the addition of sex as an independent prognostic factor in the risk-assessment tools used in the clinic, especially for patients with lower risk diseases.

## Figures and Tables

**Figure 1 cancers-13-04588-f001:**
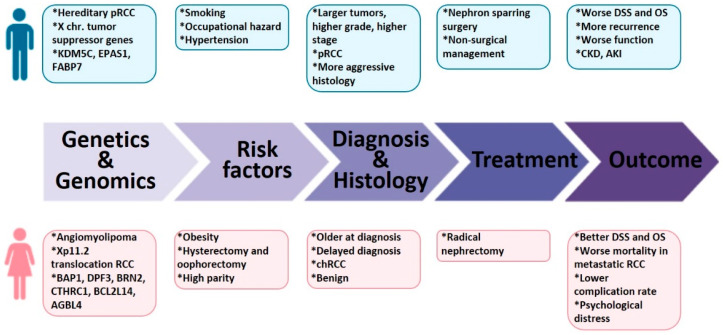
Key sex and gender-related elements to consider in clinical management of kidney cancer. RCC: renal cell carcinoma; pRCC: papillary RCC; KDM5C: lysine demethylase 5c; EPAS1: endothelial PAS domain-containing protein 1; FABP7: fatty acid-binding protein7; BAP1: BRCA1 associated protein-1; DPF3: Double PHD Fingers 3; BRN2: POU class 3 homeobox 2; ATP/GTP binding protein like 4: collagen triple helix repeat containing 1; BCL2L14: B-cell lymphoma 2-like protein 14; AGBL4: ATP/GTP binding protein like 4; chRCC: chromophobe RCC; DSS: disease-specific survival; OS: overall survival; CKD: chronic kidney disease, AKI: acute kidney disease.

**Figure 2 cancers-13-04588-f002:**
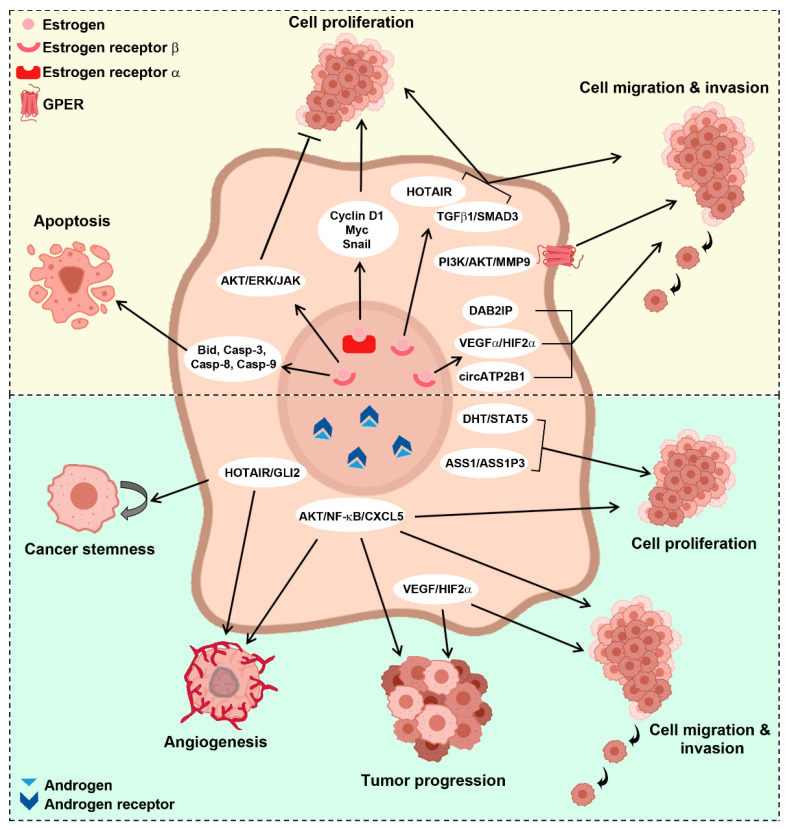
Sex hormone pathways involved in kidney cancer. AKT: protein kinase B; ERK: extracellular signal-regulated kinase; JAK: Janus kinase; TGFβ1: transforming growth factor β1; SMAD3: small mother against decapentaplegic homolog 3; DAB2IP: disabled homolog 2-interacting protein; VEGFα: vascular endothelial growth factor α; HIF2α: hypoxia-induced factor 2α; PI3K: phosphoinositide 3-kinase; GLI2: GLI Family Zinc Finger 2; ASS1: argininosuccinate synthase 1; ASS1P3: ASS1 pseudogene 3; DHT: dihydrotestosterone; STAT5: signal transducer and activator of transcription 5; GPER: G protein coupled estrogen receptor; MMP9: matrix metallopeptidase 9; HOTAIR: HOX transcript antisense RNA; NF-κB: nuclear factor kappa-light-chain-enhancer of activated B cells; CXCL5: C-X-C motif chemokine 5; Bid: BH3 interacting-domain death agonist; Casp-3, -8, -9: caspase-3, -6, -9.

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
