# Peer review of "Sex and Gender Differences in Kidney Cancer: Clinical and Experimental Evidence"

_cancers, 2021, doi:10.3390/cancers13184588_

Round 1

Reviewer 1 Report

The authors present a very interesting review paper about "Gender differences in kidney cancer: clinical and experimental evidence".

  1. Affiliations 1, 3-6, 8, 10 are the same. Please correct.
  2. I suggest expanding the “Treatment” subsection. It would be interesting to discuss the issue of how gender (and their related genes) influences the effectiveness of a specific therapeutic program for kidney cancer. I propose to collect the data in the table.

Author Response

  1. Affiliations 1, 3-6, 8, 10 are the same. Please correct.

We modified the affiliations as indicated.

  1. I suggest expanding the “Treatment” subsection. It would be interesting to discuss the issue of how gender (and their related genes) influences the effectiveness of a specific therapeutic program for kidney cancer. I propose to collect the data in the table.

We thank the Reviewer very much for his/her insightful comments and suggestion. We entirely agree with him/her that discussing how gender may influence the effectiveness of specific therapeutic interventions for patients with kidney cancer is a key research and clinical priority. However, unfortunately, high-quality studies investigating this issue are currently lacking. Most importantly, the available evidence is limited by lack of consideration of all major potential confounders (related to patients, tumours, and healthcare contexts/practices) in the analysis on the impact of sex on oncologic or functional outcomes for kidney cancer treatment (see new section lines 620-631). As such, when summarizing the currently available literature, it was challenging to highlight the concepts with a more pronounced evidence base. Consequently, we have put all our efforts to include the available studies on gender-related differences in treatment patterns and outcomes of therapy within the "Treatment" and "Outcomes" subsections of the review, respectively. While a table may provide readers with a graphical overview of the available findings, we believe it may be redundant for readers (given the low number of studies and their relatively low quality).

According to the reviewer's suggestion, we have better discussed these concepts in the revised version of the manuscript, lines 620-631. Moreover, to address a comment by Reviewer #3, we have added a paragraph in the “Treatment” subsection on the impact of sex on the outcomes of patients with metastatic kidney cancer undergoing systemic treatment with immune-checkpoint inhibitors, lines 577-614.

Reviewer 2 Report

Peired, A.J. et al. have carried out an accurate and exhaustive literature review on gender-based differences in kidney cancer, extracting data from more than 160 articles. They clearly summarized the main findings discussed in the review in Fig. 1. The authors also examined sex hormone signaling pathways involved in kidney cancer development and progression, providing a comprehensive overview of these pathways in Fig. 2.

In the final sections, the authors have emphasized the importance of taking into account gender-related specificities in the clinical management of kidney cancer patients. They also underlined the need of carrying out further basic science research and clinical research in the field of gender-based disparities in kidney cancer.

From my point of view, this is a very good review paper and could be a valuable resource for investigators in this field.

Only minor changes and corrections should be necessary, as I indicated below.

Sentences in lines 157-158 (“Shish et al. analysed the molecular signature of RCC and identified three genes (CTHRC1, BCL2L14 and AGBL4) that were differentially expressed in women”), 277-278 (“The discordant results of these studies could derive from methodological differences, and might not reflect a functional AR in RCC patients”), 310-313 (“Wang et al. reported that AR could induce RCC proliferation by regulating the miRNA availability for ASS1, through its pseudogene ASS1P3. ASSiP3 can act as a competitive endogenous RNA, altering miR-34a-5p and binding to ASS1 to suppress its expression”), and 405-406 (“Expression of progesterone receptors (PRs) in carcinomatous kidney tissue suggests a potential implication of progesterone axis in developing”) need to be rephrased more clearly and/or thoroughly.

In lines 481-487 the corresponding references must be added.

Some typographical errors should be corrected, e.g., “incressed” instead of “increased” in line 192, “ASSiP3” instead of “ASS1P3” in line 132, and “a useful” instead of “useful” in line 389.

“papillary RCC” (line 129), “clear cell RCC” (line 130), “renal cell carcinoma” (line 131), “renal cell carcinoma- ” (line 172), “hazard ratio” (line 217), and “Transforming Growth Factor Beta 1” (line 361) were not written in their abbreviated form, even if the respective abbreviations have previously been used and explained. Conversely, “RCC” first appears in line 105 without explanation of its meaning (renal cell carcinoma (RCC)).

I finally suggest to replace “despite low numbers” (lines 124-125), “expression studies” (line 137), “small sample” (line 537), and “basic science” (line 617) with “despite low participants numbers”, “gene expression studies”, “small sample size”, and “basic science research”, respectively.

Author Response

*Sentences in lines 157-158 (“Shish et al. analysed the molecular signature of RCC and identified three genes (CTHRC1, BCL2L14 and AGBL4) that were differentially expressed in women”), 277-278 (“The discordant results of these studies could derive from methodological differences, and might not reflect a functional AR in RCC patients”), 310-313 (“Wang et al. reported that AR could induce RCC proliferation by regulating the miRNA availability for ASS1, through its pseudogene ASS1P3. ASSiP3 can act as a competitive endogenous RNA, altering miR-34a-5p and binding to ASS1 to suppress its expression”), and 405-406 (“Expression of progesterone receptors (PRs) in carcinomatous kidney tissue suggests a potential implication of progesterone axis in developing”) need to be rephrased more clearly and/or thoroughly.

We thank the reviewer for his thorough lecture of the manuscript. We modified the sentences as indicated.

*In lines 481-487 the corresponding references must be added.

We added the missing reference.

*Some typographical errors should be corrected, e.g., “incressed” instead of “increased” in line 192, “ASSiP3” instead of “ASS1P3” in line 132, and “a useful” instead of “useful” in line 389.

We corrected the typos as indicated.

 *“papillary RCC” (line 129), “clear cell RCC” (line 130), “renal cell carcinoma” (line 131), “renal cell carcinoma- ” (line 172), “hazard ratio” (line 217), and “Transforming Growth Factor Beta 1” (line 361) were not written in their abbreviated form, even if the respective abbreviations have previously been used and explained. Conversely, “RCC” first appears in line 105 without explanation of its meaning (renal cell carcinoma (RCC)).

We modified the abbreviations as indicated.

*I finally suggest to replace “despite low numbers” (lines 124-125), “expression studies” (line 137), “small sample” (line 537), and “basic science” (line 617) with “despite low participants numbers”, “gene expression studies”, “small sample size”, and “basic science research”, respectively.

We modified the sentences as indicated.

Reviewer 3 Report

In this very well written review article, Peired et al. summarize sex differences in kidney cancer. The article includes most relevant literature and is scientifically sound. I suggest to address the following comments to get a complete picture:

  1. Please add a section about sex differences with regard to inflammation. This is especially relevant because PD-1/PD-L1  immune checkpoint inhibition is first-line in these patients. 
  2. Please discuss current therapeutical concepts, including immune checkpoint inhibitor and tyrosine kinase inhibitor therapies.
  3. I suggest to replace the term gender to sex since the article refers to biological rather than social conditions. This is equal for using male/female rather than women/men. 

Author Response

  1. Please add a section about sex differences with regard to inflammation. This is especially relevant because PD-1/PD-L1  immune checkpoint inhibition is first-line in these patients. 

A paragraph summarizing the sex differences with regard to inflammation in kidney cancer patient was already present line 148-153 and 158-164. To the best of our knowledge, no additional article exists on the topic. Following the reviewer suggestion, we underlined the importance of inflammation in kidney cancer development lines 165-175, and developed the use of checkpoint inhibition in kidney cancer treatment in the 2”Treatment” section of the manuscript, lines 577-614.

  1. Please discuss current therapeutical concepts, including immune checkpoint inhibitor and tyrosine kinase inhibitor therapies.

We thank the Reviewer very much for his/her critical comments. According to the Reviewer's suggestion, we have expanded the "Treatment" section of the review by highlighting the contemporary role of systemic therapy with immune-checkpoint inhibitors for metastatic kidney cancer lines 577-614, as well as the (lack of) available data on gender-related differences in oncologic outcomes in such patients, lines 620-631.

  1. I suggest to replace the term gender to sex since the article refers to biological rather than social conditions. This is equal for using male/female rather than women/men. 

We thank the reviewer for raising this important point. We reviewed our use of gender in the whole manuscript and replaced it by sex every time it was necessary. We also replaced women/men with male/female. We also added an introductory sentence to the “Introduction” section, lines 55-59.

Reviewer 4 Report

The main issue is the differences in the development of kidney cancer among men and women. The authors try to justify the differences in the incidence of kidney cancer. The existing gender differences are widely known, but their nature remains unexplored to date.

The section on the presence of genetic factors capable of determining such a picture of the disease is quite interesting. Of course, differences in hormonal levels and the presence of androgen and estrogen receptors are key factors. I would like to note that such facts have been known for a long time. However, they are not taken into account by clinical oncologists. Probably, the molecular picture of the development of kidney cancer in men and women determines not only the prognosis of the disease, the response to anticancer therapy, but can also be used to present new approaches for finding targets for personalized therapy.

The article is included in the sphere of interest of the journal. It is not only epidemiological data but also genomic and molecular features. Since from a scientific point of view, the authors will substantiate the differences in the incidence of kidney cancer among men and women.

There have been no comprehensive studies in the literature trying to bring together various factors in the pathogenesis of kidney cancer.

The merit of the article is the consideration of various factors in the development of kidney cancer. Of particular interest is the article's structure, which shows the authors' train of thought and logically follows from the presented material. The conclusions are consistent and follow from the material

The references are appropriate. The Figures and tables are attractive, adding additional information to the paper.

Author Response

We thank the reviewers for his/her detailed analysis of the manuscript, and by extension of the relevance of sex and gender in kidney cancer, in particular for the clinical management of the patients.